# Immunotherapeutic Potential of the Yellow Fever Virus Vaccine Strain 17D for Intratumoral Therapy in a Murine Model of Pancreatic Cancer

**DOI:** 10.3390/vaccines13010040

**Published:** 2025-01-06

**Authors:** Alina S. Nazarenko, Yulia K. Biryukova, Kirill N. Trachuk, Ekaterina A. Orlova, Mikhail F. Vorovitch, Nikolay B. Pestov, Nick A. Barlev, Anna I. Levaniuk, Ilya V. Gordeychuk, Alexander S. Lunin, Grigory A. Demyashkin, Petr V. Shegai, Andrei D. Kaprin, Aydar A. Ishmukhametov, Nadezhda M. Kolyasnikova

**Affiliations:** 1Laboratory of Tick-Borne Encephalitis and Other Viral Encephalitides, Chumakov Federal Scientific Center for Research and Development of Immune-and-Biological Products of RAS (Institute of Poliomyelitis), Moscow 108819, Russiakolyasnikova_nm@chumakovs.su (N.M.K.); 2Department of Organization and Technology of Production of Immunobiological Preparations, Institute for Translational Medicine and Biotechnology, I.M. Sechenov First Moscow State Medical University (Sechenov University), Moscow 117418, Russia; gordeychuk_iv@chumakovs.su; 3Vavilov Institute of General Genetics, Gubkina 3, Moscow 119991, Russia; 4Laboratory for Modelling Immunobiological Processes with Experimental Clinic of Callitrichidae, Chumakov Federal Scientific Center for Research and Development of Immune-and-Biological Products of RAS (Institute of Poliomyelitis), Moscow 108819, Russia; 5Laboratory of Histology and Immunohistochemistry, Institute for Translational Medicine and Biotechnology, I.M. Sechenov First Moscow State Medical University (Sechenov University), Moscow 117418, Russia; 6Department of Digital Oncomorphology, National Medical Research Centre of Radiology, 2nd Botkinsky pass 3, Moscow 125284, Russia; 7Department of Urology and Operative Nephrology, Peoples’ Friendship University of Russia (RUDN University), Miklukho-Maklaya str.6, Moscow 117198, Russia

**Keywords:** virotherapy, cancer immunotherapy, 17D, yellow fever vaccine, intratumoral administration, PDAC, pancreatic cancer, Pan02

## Abstract

**Objective**: We evaluate the immunotherapeutic potential of the yellow fever virus vaccine strain 17D (YFV 17D) for intratumoral therapy of pancreatic cancer in mice. **Methods**: The cytopathic effect of YFV 17D on mouse syngeneic pancreatic cancers cells were studied both in vitro and in vivo and on human pancreatic cancers cells in vitro. **Results**: YFV 17D demonstrated a strong cytopathic effect against human cancer cells in vitro. Although YFV 17D did not exhibit a lytic effect against Pan02 mouse cells in vitro, a single intratumoral administration of 17D caused a delay in tumor growth and an increase in median survival by 30%. Multiple injections of 17D did not further improve the effect on tumor growth; however, it notably extended the median survival. Furthermore, preliminary immunization with 17D enhanced its oncotherapeutic effect. **Conclusions**: Intratumoral administration of yellow fever virus vaccine strain 17D delayed tumor in a murine model of pancreatic cancer. The fact that YFV 17D in vitro affected human cancer cells much more strongly than mouse cancer cells appears promising. Hence, we anticipate that the in vivo efficacy of YFV-17D-based oncolytic therapy will also be higher against human pancreatic carcinomas compared to its effect on the mouse pancreatic tumor.

## 1. Introduction

Pancreatic cancer ranks as the ninth most commonly diagnosed malignant tumor and is one of the deadliest cancers, with a 5-year survival rate of around 5–8% following diagnosis [1,2]. The majority of cases are adenocarcinomas, originating from acinar cells and pancreatic ductal glandular cells–PDAC (pancreatic ductal adenocarcinoma). PDAC cells are characterized by specific somatic mutations in oncogenes KRAS (90%), CDKN2 (80%), p16 (75%), p53 (70%), and SMAD4 (65%) [3,4]. If diagnosed late, the majority of PDAC patients are burdened with inoperable, often metastatic disease. The asymptomatic progression and the onset of specific symptoms only at advanced stages of the disease contribute to challenges in early detection of PDAC, ultimately affecting survival outcomes. For patients with operable tumors, treatment options include systemic chemotherapy (e.g., gemcitabine and paclitaxel) or combination chemoradiotherapy. In cases of locally advanced, non-resectable PDAC, treatment typically includes polychemotherapy (e.g., FOLFIRINOX) or intensive chemoradiotherapy [5]. Given the limited efficacy of these therapies for patients with PDAC, numerous new treatment approaches are under investigation in clinical trials. These include immune checkpoint inhibitors [6], inhibitors of polyamine biosynthesis with the simultaneous blockade of polyamine transport [7], microrobots [8], and various immunotherapeutic agents, such as anti-mesothelin CAR-T cells (for example, NCT01897415) [9] and oncolytic viruses [10].

Oncolytic viruses (OVs) represent a novel class of cancer immunotherapy agents that preferentially infect and kill cancer cells and promote protective antitumor immunity [11]. The most frequently studied in clinical trials OV platforms for pancreatic cancer include reoviruses, herpes simplex viruses, vesicular stomatitis viruses, parvoviruses, and adenoviruses [12]. The local destruction of tumor tissue by oncolytic viruses (OVs) triggers the activation and maturation of dendritic cells and tumor-specific T cells through the cross-presentation of tumor antigens, leading to a transformation of the tumor immune microenvironment into an immunologically active (“hot”) state. The immune resistance characteristic of PDAC significantly diminishes the efficacy of current immunotherapies based on immune checkpoint inhibitors (e.g., anti-CTLA-4, anti-PD-1). However, clinical trial data indicate that combining PD-1-blocking antibodies, chemotherapy, and OVs can enhance the antitumor efficacy in pancreatic cancer [13].

The search for more effective OVs is ongoing. The oncolytic activities of flaviviruses are being extensively studied in experimental animal models across various tumor types. In a previous review, we summarized these findings and evaluated the potential of flaviviruses for antitumor therapy [14]. In terms of immunogenicity, flaviviruses may positively impact tumor therapy and promote oncolysis through an immune-mediated pathway. Aznar et al. [15] demonstrated that the live-attenuated 17D vaccine strain of the YFV effectively infects and induces cell death in both murine and human tumor cells from different tissues of origin (representing colon cancer, renal cell carcinoma, breast cancer, and melanoma), with the multiplicity of infection (MOI) as low as 10, which is innocuous for non-transformed human fibroblasts. Intratumoral administration of 17D inhibits MC38 and B16OVA tumor progression in C57BL/6 mice. McAllister et al. showed that a recombinant live-attenuated 17D YFV strain bearing a chicken ovalbumin epitope (SIINFEKL), which is recognized by cytotoxic T lymphocytes, serves as an effective therapeutic vaccine for the treatment of murine experimental solid tumors with pulmonary metastases of B16-OVA murine melanoma cells expressing chicken ovalbumin [16].

Therefore, 17D YFV can be considered as a promising antitumor agent, and its status as a widely used yellow fever vaccine for human prophylaxis significantly simplifies the clinical development of immunotherapy strategies utilizing intratumoral injections [17,18,19].

Pan02 cells represent a syngeneic model of PDAC in C57BL/6 mice, popular in preclinical studies. The tumor growth kinetics of subcutaneously implanted Pan02 cells allow for a more than three-week therapeutic window to assess the antitumor effects of investigational treatments [20].

In this study, we demonstrate the immunotherapeutic potential of 17D YFV for intratumoral therapy in vivo in a syngeneic murine model of pancreatic cancer. We evaluate the efficacy of preliminary 17D immunization and multiple intratumoral administrations of 17D, observing delayed tumor growth, enhanced survival, and the histological manifestations. Additionally, we show that 17D YFV acts effectively as an OV virus against human pancreatic cancer cells in vitro.

## 2. Materials and Methods

### 2.1. Cell Lines

Vero monkey kidney epithelial cells (ATCC CCL-81) were obtained from the American Cell Culture Collection (ATCC), and the human pancreatic carcinoma cell lines MIA PaCa-2, pancreas ductal epithelioid carcinoma PANC-1, and murine pancreatic ductal adenocarcinoma Pan02 cells were from the collection of the Laboratory of Tick-Borne Encephalitis and Other Viral Encephalitides (Chumakov FSC R&D IBP RAS (Institute of Poliomyelitis), Moscow, Russia).

Cells were cultured in DMEM nutrient growth medium (Chumakov FSC R&D IBP RAS (Institute of Poliomyelitis)), with the addition of 2 mM L-glutamine (Paneco, cat. no. F032, Moscow, Russia), antibiotics penicillin (250 IU/mL) and streptomycin (200 μg/mL) (Paneco, cat. no. A065p), and 5% fetal bovine serum (FBS Gibco #2412072), and they were incubated in an atmosphere of 5% CO_2_ at 37 °C. After the infection of the cells, we used IMEM (Chumakov FSC R&D IBP RAS (Institute of Poliomyelitis)) as a supporting medium with additives identical to those in DMEM but with 2% FBS.

All of the cell lines used in this study tested negative for mycoplasma (mycoplasma detection kit Myco Real-Time, Evrogen, cat. No. MR004, Moscow, Russia).

### 2.2. Cultivation and Quantification of 17D YFV

We used the attenuated vaccine strain 17D YFV from the collection of Chumakov FSC R&D IBP RAS (Institute of Poliomyelitis). To produce viral stocks, Vero cells were infected with 2.5 × 10^6^ PFU/mL 17D YFV. The supernatant was collected on the third day after infection, when the cytopathic effect (CPE) exceeded 80%, and was purified from cell debris by centrifugation at 3000× *g* for 30 min at 4 °C. The virus stock was stored at −80 °C. The virus titer was assessed by plaque assay. Serial ten-fold dilutions of 17D YFV were inoculated into a monolayer of Vero cells in 12-well plates and incubated for 60 min in an atmosphere of 5% CO_2_ at 37 °C. After, a 1 mL coating consisting of medium 199 with Earle and Hanks salts (Chumakov FSC R&D IBP RAS (Institute of Poliomyelitis)) with 1.25% methylcellulose (Merc), 0.1% gentamicin (Paneco), and 2% FBS was added at a ratio of 2:1. Seven days post-infection, the coating was removed and the cells were washed with PBS, fixed with 95% ethanol, and stained with 0.1% crystal violet dye (Sigma-Aldrich, Burlington, MA, USA). The number of plaques for each dilution were then counted to estimate the plaque-forming units per mL (PFU/mL).

To determine the titer of infectious 17D virus in the removed tumors, tumor exudates that had previously been analyzed by PCR were used to infect the Vero cells, followed by incubation for 5 days, after which the culture fluid was collected and analyzed by PCR for the presence of 17D.

### 2.3. RNA Extraction and Quantitative RT-PCR

The concentration of viral RNA in the culture medium was assessed using the quantitative PCR method. A 1 mL sample of the culture medium was used, and the cell debris was removed by centrifugation at 3000 rpm for 5 min. Isolation of viral RNA from the supernatant was carried out using the Ribo-prep reagent kit. PCR was carried out using the AmpliSens YFV-FL reagent kit, and YFV strain 17D was used as the standard (calibrator) at titers of 1 × 10^7^, 1 × 10^5^, and 1 × 10^3^ PFU/mL. The manufacturer of all of the reagent kits used for the RNA was the Central Research Institute of Epidemiology of Rospotrebnadzor (Moscow, Russia). Quantitative real-time PCR was performed on a Rotor-Gene Q6plex thermal cycler (Qiagen) using detection channels FAM and HEX.

The concentration of viral RNA in the tumor samples taken from euthanized mice was determined by quantitative real-time PCR—genome equivalents (GE/mL). The tumor fragments were ground with a mortar and pestle in PBS, centrifuged, and then the RNA was isolated and RT-PCR of viral RNA was performed using the Ribo-prep and AmpliSens YFV-FL reagent kits as above.

### 2.4. In Vitro Sensitivity Estimation Using Cell Viability Assays

Cell viability was assessed with the MTT assay. Ninety-six-well plates holding the cells of the transplantable tumor lines were infected with 10-fold serial dilutions of the virus in six repetitions. Viral dilutions were added at a volume of 30 μL per well; incubation of the virus with cells was carried out for 1 h in an atmosphere of 5% CO_2_ at 37 °C. Then, the virus-containing liquid was removed and the cells were washed with a serum-free medium and cultured in a maintenance medium with 2% FBS. After 3, 4, and 5 days, the cell viability was determined with the MTT reagent at a working concentration of 5 mg/mL in phosphate buffer pH 7.2 (Paneco, cat. no. O104). The optical density was determined as the difference in the optical densities at 595 and 650 nm using a Bio-Rad iMark plate reader. The result was evaluated in comparison with the average optical density in the control (uninfected cells), taking it as 100%.

### 2.5. xCELLigence Real-Time Cell Proliferation Measurement

Cell proliferation was monitored using Agilent’s Real-Time Cell Analysis (RTCA) system. For determination of the proliferation curves, E-Plates were seeded with 10^4^ cells/well in the appropriate growth medium and cultured in an atmosphere of 5% CO_2_ at 37 °C. Signals were measured every 2 h on the first day and every 4 h thereafter. Data were analyzed using RTCA Pro Software 2.6.0 (Basic), which generated the curve for the wells (n = 8) with standard deviations (SD).

To study the CPEs (cytopathic effects) of the virus on cells, the cells were mixed with 17D in the growth medium at MOI = 100, and 100 µL of 10^4^ cells were placed in each well of the E-Plate. Cell proliferation was monitored over 120 h. The control was uninfected cells subjected to the same manipulations.

Cells were infected with 10^6^ PFU/mL 17D in a 25 cm^2^ flask. Three days post-infection, cells were seeded at 1.5 × 10^5^ cell/well on E-plates for the xCELLigence Real-Time cell proliferation measurement. The medium was replaced after 48 and 96 h. The control was uninfected cells subjected to the same manipulations.

The cells were allowed to achieve approximately 70% confluency over 24 h and then were infected with 17D (MOI = 1) directly in an E-plate. After 1 h of exposure, the cells were washed and cultured in an atmosphere of 5% CO_2_ at 37 °C. The cell proliferation was monitored over 144 h. The control was uninfected cells subjected to the same manipulations.

### 2.6. Analysis of YFV Fusion with PANC-1, MIA PaCa-2, and Pan02 Cells

To obtain DiD-labeled virions, the pre-clarified virus-containing liquid was ultracentrifuged twice in 20% sucrose in PBS and then in pure PBS at 25,000 rpm at 4 °C for 2.5 h to purify the virus. Then, 1 μL of a 0.1 mM solution of 1,1′-dioctadecyl-3,3,3′,3′-tetramethylindicarbocyanine (LumiTrace DiD, Lumiprobe, Moscow, Russia) in DMSO was mixed with 10 μL of purified YFV (10^6^ TCID_50_/_mL_) in PBS. After pipetting, 305 μL of FluoroBrite DMEM (Gibco, Thermo Fisher Scientific) was added to the mixture. A suspension of DiD-labeled YFV virions inactivated with 0.01% formaldehyde was used as a negative infection control. Unreacted DID was removed using a centrifugal concentrator at 7000 rpm for 10 min. After mixing with a pipette, the resulting mixtures were added to PANC-1, MIA PaCa-2, and Pan02 cells at 60 μL per well in duplicates and then were incubated in the atmosphere of 5% CO_2_ at 37 °C. After 1.5 h, the cells were visualized using an OLYMPUS CKX53 fluorescence inverted microscope with 20× magnification, Cy5 filter cube: Ex620/60, Em700.

### 2.7. 17D Intratumoral Administration and the In Vivo Efficacy Experiments

Six- to eight-week-old female C57BL/6 mice were purchased from the SPF-vivarium of the Institute of Cytology and Genetics SB RAS, Novosibirsk, Russia (https://ckp.icgen.ru/, accessed on 26 December 2024). All animal procedures were carried out strictly in accordance with the Protocol No. 210223 approved by the Ethics Committee of Chumakov Federal Scientific Center for Research and Development of Immune-and-Biological Products. Pan02 cells (5.2 × 10^5^ cells per 100 μL of PBS) were implanted subcutaneously into the right flank of the mice. The condition of the mice was monitored daily. Blood sampling to assess hematological parameters (WBCs, LYM, MID, GRAN, RBC, HGB, and PLT) was carried out four times.

Tumors were measured twice weekly using a caliper. All tumor measurements were performed in duplicate or triplicate. The tumor volume was calculated using the following formula:$$V=\frac{a*b^{2}}{2},$$
where

*a* is the larger of the two dimensions;

*b* is the smaller of the two dimensions.

When the tumor reached a size of 5–7 mm in diameter (100 mm^3^), intratumoral therapy was initiated (day 0) by injecting 100 μL of 17D (DMEM Vero) with a titer of 10^6^ PFU/100 μL. The mice from the control group were intratumorally injected with 100 μL of DMEM culture medium, which was obtained from flasks with growing Vero cells. In the group of animals with triple 17D virotherapy, 10^6^ PFU/100 μL 17D intratumoral injections were administered three times every seven days.

Upon reaching a humane endpoint—defined as a tumor greater than 10 mm in two dimensions or a significant decline in health of the animal, potentially causing intense suffering (limb atrophy, starvation, and critical weight loss)—mice were euthanized, necropsied, and their blood and tumor samples were collected. The tumors, along with nearby lymph nodes, were excised and fixed in formaldehyde for histological examination. The efficacy of the therapy was evaluated based on tumor growth kinetics, overall health status of the mice (including injection tolerance, complete blood count, weight changes, and behavioral assessment), and survival time.

To evaluate the abscopal effect due to the systemic antitumor effect of the virus, groups of mice were subcutaneously implanted into both thighs with 600 thousand Pan02 cells in 100 μL. When one of the tumors reached a size of 3–5 mm, an intratumoral injection of 17D was administered in only one tumor, located on the side of the right thigh, at a dose of 10^6^ PFU/100 μL. In the control group, 100 μL of DMEM culture medium from the Vero cell line was injected into one tumor on the right side.

The number of mice in the experimental groups was n = 10, which was reduced to n = 8 before starting therapy according to the following exclusion criterion: tumor size less than 4 mm or more than 7 mm within the therapeutic window of one week.

### 2.8. 17D Immunization and Antibody Neutralization Assay

Immunization of mice against yellow fever virus was carried out with a single subcutaneous injection (at the base of the tail) using the 17D strain (Chumakov FSC R&D IBP RAS (Institute of Poliomyelitis)), the virus was passaged on Vero, virus titer of 10^6^ PFU/mL) using 100 µL per mouse. To evaluate the effectiveness of the immunization, a neutralization assay was performed 14 days later using sera from immunized mice.

### 2.9. Neutralization Reaction Analysis

A plaque assay was used to analyze 17D-neutralizing antibodies titers in mice. Sera from the blood of the 17D-immunized mice were collected on day 14 after immunization and at the end of the experiment. The sera were then diluted in serial three-fold dilutions and mixed with 17D (3 × 10^3^ PFU/mL), followed by incubation for 1 h at 37 °C. 17D (1.5 × 10^3^ PFU/mL) was used as a control. The resulting mixtures were used to infect a monolayer of Vero cells in 12-well plates with incubation for 60 min in an atmosphere of 5% CO_2_ at 37 °C. After, a 1 mL coating, consisting of medium 199 with Earle and Hanks salts (Chumakov FSC R&D IBP RAS (Institute of Poliomyelitis), Moscow, Russia) at a ratio of 2:1, 1.25% methylcellulose (Merck), 0.1% gentamicin (Paneco), and 2% FBS, was added. Seven days post-infection, the coating was removed and the cells were washed with PBS, fixed with 95% ethanol, and stained with 0.1% crystal violet dye (Sigma-Aldrich). The number of plaques for each dilution was then counted to calculate the neutralizing antibody titer. The calculation was made in logarithmic units using the following formula:lg(nAB) = lg(A) + (f ∗ lg(d)),
where

A—dilution at which the number of plaques is less than 50% compared to control;

f—logarithm difference (interpolation coefficient), calculated as follows: f = (50% − M1)/(M2 − M1), where

M1—number of plaques is less than 50% compared to the control;

M2—number of plaques in the next dilution (plaques > 50%);

d—dilution factor.

### 2.10. Statistical Analysis

Prism 8 software (GraphPad Prism Software, Inc., La Jolla, CA, USA) was used for the data analysis. The two-tailed Student’s *t*-test or Mann–Whitney test was used to analyze statistical differences between the two groups. The Kruskal–Wallis test was used to analyze statistical differences among three or more groups. Statistical significance for the Kaplan–Meier survival curves was determined using the log-rank test (Mantel–Cox test). *p*-Values of *p* < 0.05 (*), *p* < 0.01 (**), and *p* < 0.001 (***) were considered statistically significant.

## 3. Results

### 3.1. Cytopathic Effects of the 17D Virus on Pancreatic Carcinoma Cell Lines In Vitro

We investigated the in vitro cytopathic effect (CPE) of 17D on human PANC-1, MIA PaCa-2, and mouse Pan02 pancreatic carcinoma cell lines. Cells were exposed to increasing 17D virus titers in culture from 10^2^ to 10^7^ PFU/mL. Cell viability was assessed by the MTT assay performed after 3 and 5 days. It was found that 17D had an evident oncolytic effect on MIA PaCa-2. After five days of exposure to 10^3^ PFU/mL 17D, cell viability was >50%, whereas 10^6−7^ PFU/mL 17D resulted in zero cell viability (Figure 1A). PANC-1 cells were more resistant to 17D infection, showing approximately 50% survival at 10^6^ PFU/mL 17D exposure (Figure 1B). In contrast, Pan02 cells demonstrated remarkably low susceptibility to the 17D infection (Figure 1C).

### 3.2. Evaluation of the Persistence of 17D in Pan02 Cells In Vitro

The weak CPE of 17D on the Pan02 cells prompted us to increase the multiplicity of infection (MOI). Thus, at a high MOI = 100, the Pan02 cells’ proliferation was inhibited, but—as opposed to the human cell lines, MIA PaCa2 and PANC-1—this inhibition was incomplete (Figure 1D,E). Some resistant Pan02 cells were able to recover and form a new monolayer after a medium change, indicating that CPE of 17D in Pan02 cells is have not resulted from their direct lysis but from growth inhibition.

Microscopy of the Pan02 cells, both infected and uninfected, after three days of exposure to 17D infection revealed notable differences in monolayer formation. Upon prolonged incubation in a regularly changed medium, the uninfected Pan02 cells maintained cell–cell contact and exhibited multilayer growth. In contrast, the 17D-infected Pan02 cells displayed reduced capacity to form a second layer, also showing an increased number of detached cells (Figure 2A,D). Quantification of the detached cells (representing dead cells) in the culture medium after three days showed a significantly higher number in the 17D-infected cultures, at 4.02 × 10^4^ cells/mL for infected compared to 5.74 × 10^3^ cells/mL for uninfected (i.e., control cell) cultures, despite the intact cell monolayer. Real-time observation of Pan02 cell proliferation following 17D infection demonstrated normal cell growth within the initial 36 h. However, by 48 h post-infection, a noticeable decrease in the proliferative activity of infected cells was observed without evident monolayer degradation (Figure 2B,D). This effect might be attributed to the binding of virions to the surface membrane receptors without subsequent receptor-mediated endocytosis (fusion step) and replication of the virus.

To answer this question, we looked for evidence of the presence of viral RNA in the media and cells, as well as the ability of 17D to infect Pan02 cells compared to human cell lines. Replacing the culture medium led to the disappearance of the 17D-induced CPE in the Pan02 cells but did not eliminate the viral RNA. While the proliferation of 17D-infected Pan02 cells was initially inhibited, replacement of the culture medium at 48 h post-infection resulted in the growth curve leveling off by 72 h (Figure 2B). The initial 17D infection dose titer was 10^5^ GE/mL (according to the RNA concentration as determined by RT-PCR) and remained at the same level (4.6 × 10^5^ GE/mL) after three days, decreasing to 8 × 10^3^ GE/mL after 48 h. After 168 h of culturing, replacing the maintenance medium twice, viral RNA was still detectable at 4.4 × 10^4^ GE/mL (Figure 2B,C). Moreover, viral RNA persisted both in the culture medium and within the cells after thirteen days of Pan02 cultivation (Table 1). Furthermore, the 17D viral RNA and plaque-forming units were detected in Pan02 cultures despite three subsequent passages (Table 2).

In contrast to the mouse Pan02 cells, the 17D viral RNA levels in the culture media of the human carcinoma cell lines were remarkably higher. From the initial concentration of 17D = 1 × 10^6^ GE/mL, it increased after 3 days to 10^7^ GE/mL for the PANC-1 cells and to 10^8^ GE/mL for the MIA PaCa-2 cells (Table 3). The increased sensitivity of human carcinoma cells to 17D allows for the use of lower levels of multiplicities of infection (MOIs) without compromising oncolytic efficacy, although it results in a delay in the effect. For instance, MIA PaCa-2 cells infected with 17D at an MOI = 1 exhibited proliferation curves similar to those of uninfected cells for the first 48 h, followed by complete cell lysis (Figure 2E).

### 3.3. 17D Infection of Pan02 Cells

To confirm the viral entry into cells, we employed the lipophilic membrane tracer DiD, which enhances fluorescence when viral capsid fuses with the cell membrane [21,22,23,24] (visible in the Cy5 channel). Pan02, MIA PaCa-2, and PANC-1 cells were infected with live DID-labeled 17D virus, while the DiD-labeled formalin-inactivated 17D virus served as a negative control. After 2 h of incubation, fluorescent signals indicating membrane fusion events were observed for the infectious virus, whereas the inactivated virus produced only a weak signal (Figure 3). The fluorescence intensity indicating 17D fusion was lower in Pan02 cells compared to human MIA PaCa-2 and PANC-1 cells.

### 3.4. Intratumoral Administration of 17D Inhibits Tumor Growth and Improves Survival in a Murine Model of Pancreatic Cancer

After subcutaneous implantation of 5.2 × 10^5^ Pan02 cells in 100 μL PBS in the right flank of C57BL/6 mice, the tumors reached a size of 5–7 mm by day 10. At this point, 10^6^ PFU/100 μL of 17D was administered intratumorally for the virotherapy group, and 100 μL DMEM culture medium from uninfected Vero cells was administered intratumorally for the control group (Figure 4A). Mice were euthanized upon reaching the humane endpoint after which autopsies were performed and the blood and tumor tissues were collected.

Intratumoral 17D administration resulted in a delayed tumor growth and an increased median survival by 30%. The maximum survival time of the mice in the 17D-treated group reached 42 days versus 25 days in the control group (Figure 4B,C). A single 17D intratumoral injection was well tolerated by the mice with no significant changes in health status observed compared to the control group. Complete blood count parameters stayed within the reference values (according to the Charles River breeding center standards [25]), and no weight loss was observed until the last week preceding the humane endpoint. In the virotherapy group, we observed edema at the tumor sites during the first 7 days post-injection in some mice, complicating the exact measurement of the tumor size during this period (Figure 4C).

Histological analysis of the tumors, collected from the control group of the mice, euthanized on day 25, revealed that the adenocarcinoma parenchyma consisted of densely packed cells interspersed with loosely arranged, atypical cells primarily of the epithelioid (glandular) type, where mitotic figures and dead single cells, putatively by apoptosis, were also visible. Notably, in the control group, regional lymph nodes were not enlarged and no metastases were detected. In the virotherapy group, tumor fragments from mice at 25, 35, and 42 days were analyzed. The tumors exhibited a solid structure with active proliferation of atypical cells, focal necrosis, and diffusely scattered lymphocytes. No distinct progressive pathomorphological changes were noted. Tumor growth appeared to stabilize. There were no histological differences in lymph nodes between the two groups, and no metastases were detected (Figure 4D,E).

### 3.5. 17D Pre-Immunization Enhances the Efficacy of Intratumoral 17D Therapy

We aimed to enhance the antitumor effects through activation of immune mechanisms. Previously, Aznar et al. [15] showed that 17D possessed enhanced antitumor effects after a pre-immunization in the MC38 and B16-OVA subcutaneous tumor models, and this effect might attributed to CD8+ T cells. Thus, we immunized the mice with 10^6^ PFU/100 μL 17D by subcutaneous injection at the base of the tail. The sera from the 17D-immunized mice were collected at day 14 post-immunization, and neutralizing anti-17D antibodies were detected in all of these mice (Table 4). Then, after 4 days, Pan02 cells were implanted subcutaneously. The tumors reached a size of 5–7 mm on day 10 post-implantation and then 10^6^ PFU/100 μL 17D was administered intratumorally (for virotherapy group) and 100 μL DMEM culture medium from uninfected Vero cells was administered intratumorally for the control group (Figure 5A).

In terms of growth kinetics, 17D therapy had a more pronounced positive effect in immunized mice, resulting in a 56% increase in median survival (Figure 5B,C). The mice tolerated both the immunization and subsequent single intratumoral 17D injection without any significant changes in health status compared to the control group. Complete blood count parameters remained within reference ranges, and no weight loss was observed until the final week prior to reaching the humane endpoint.

Histological examination of tumors collected from immunized mice that received the virotherapy and were euthanized on day 25 of the experiment revealed moderate inflammatory mononuclear infiltration in the peritumoral zone, as well as foci of cell death of atypical cells. These changes were not observed in the control group.

In contrast, mice from the virotherapy group euthanized on day 35 showed less prominent inflammatory mononuclear infiltration and larger foci of cell death (Figure 5D). The tumors were predominantly of a trabecular–fascicular structure, in contrast to solid tumors observed in non-immunized mice treated with the virotherapy. The lymph nodes in the virotherapy-treated mice showed no differences from those of the control group.

### 3.6. Multiple Intratumoral Administrations of 17D

In the group of non-immunized mice with of 10^6^ PFU/100 μL of 17D injected intratumorally once per week for three weeks (i.e., triple virotherapy), the median survival was not significantly improved as compared to the single virotherapy group. The overall median survival in the group of non-immunized mice with triple virotherapy increased by 15% (Figure 6). The median survival in the group of immunized mice treated with the triple virotherapy was 43% higher compared to the control group. Remarkably, the median survival in the group of immunized mice treated with a single dose of 17D was by 56% higher compared to the control group. However, one mouse from the group of immunized mice treated with the triple virotherapy lived for 49 days, which exceeded the median survival in the group of immunized mice treated with a single dose of 17D (Figure 7).

After the third injection of 17D, some mice exhibited evident signs of decline in health, such as reduced motor activity and periods of unconsciousness. These signs resolved within 24 h. Peritumoral edema was also observed in this group of mice. Complete blood count parameters remained within reference ranges before euthanasia, and no body weight loss was detected.

According to the histological examination, the immunological effects of the multiple virotherapy were stronger than those of the single-dose regimen. This was likely due to a more pronounced inflammatory response and increased tumor infiltration by immune cells. However, the resulting peritumoral edema, rather than the tumor growth, might have contributed to the mice reaching the humane endpoint earlier (Figure 7D,E).

PCR analysis of tumor fragments collected at the end of the study did not show the presence of the 17D viral RNA. Five tumor and serum samples from the 17D-treated mice euthanized on days 32, 36, 38, and 50 were analyzed and tested negative for 17D. Additionally, no infectious virus was recovered upon exposure of tumor exudates to Vero cells. This might indicate the rapid elimination of 17D after the injection.

### 3.7. Delayed Contralateral Tumor Growth in the Immunized Mice

Despite the fact of rapid 17D elimination in C57BL/6 mice, administration of 17D intratumorally still gives an advantage to the immunized mice. Moreover, this advantage was retained in the mice with bilateral tumors (Figure 8). The results showed that the virotherapy-treated tumors grew more slowly than contralateral tumors without the virotherapy. Additionally, the contralateral tumors showed delayed growth, and the overall survival was increased by 59.5% compared with the control group. This effect might be attributed to immune response rather than the direct viral infection of the non-treated contralateral tumors.

### 3.8. Viral Load, Neutralizing Antibodies, and White Blood Cells Analyses

Mice were immunized with 10^6^ PFU/mL of 17D by subcutaneous injection at the base of the tail with the subsequent virotherapy of subcutaneous tumors at the same dose. The increased number of injections (×3) did not improve the effectiveness of the therapy. Each subsequent viral load caused an increase in the level of neutralizing antibodies in the sera of the 17D-immunized mice (Figure 9). The highest antibody titer against 17D at the humane endpoint was detected in the group with triple virotherapy compared to the control group, where the titer started to decrease. Thus, we increased the resistance to 17D with each subsequent injection. This could have been avoided by increasing the dose of the virus with each subsequent injection.

During the virotherapy, hematological parameters (WBCs, LYM, MID, GRAN, RBC, HGB, and PLT) of C57BL/6 mice were monitored at different blood sampling points: 0—intact mice (experiment start); 1—day 14 post-immunization; 2—after Pan02 implantation prior to 17D virotherapy; and 3—humane endpoint (experiment end). Complete blood count parameters stayed within reference values. This means that the mice tolerated all manipulations and virotherapy until the humane endpoint, when the tumors grew up to 1000 mm^3^ (Figure 10A,B). However, if we consider each case individually for each mouse, some peculiarities are present. Analyzing each mouse, we observed that one mouse responded to the therapy most effectively (49 survival days). By contrast, in a mouse from the control group with the most aggressively growing tumor (22 survival days), we found some changes in the WBC parameters. The long-lived mouse proved to be more immunoreactive; its response to tumor cell implantation correlated with an increased level of WBCs (Figure 10C). The greatest contributors to the WBC population were LYM 80.92%, MID 2.55%, and GRAN 16.53%. The evaluation of the contribution of a specific T-lymphocyte subpopulation for 17D therapy was not performed in this study.

## 4. Discussion

Our study demonstrates that 17D YFV possesses an immunotherapeutic potential for the growth retardation of subcutaneous Pan02 tumors. Since 17D showed no lytic effect on Pan02 cells in vitro, we did not anticipate a complete tumor regression. Nevertheless, intratumoral injection of 17D resulted in a delayed tumor growth and up to 30% increase in median survival. Additionally, while multiple virus injections did not inhibit tumor growth further, this improved overall survival. Notably, preliminary 17D immunization substantially enhanced the therapeutic effect, with a 56% increase in median survival.

We propose that these findings are promising, especially because the human pancreatic carcinoma cells showed a more pronounced in vitro CPE with the 17D virus. However, we recognize the risk that our expectations may not be met due to a number of factors, including both mechanisms of cell resistance that impair viral tropism.

The higher resistance of PANC-1 cells to 17D infection, compared to MIA PaCa-2 cells, may be attributed to the increased expression of IRF2 and IRF7, which regulate the interferon α and γ genes in this cell line [26,27].

Predicting the therapeutic efficacy of oncolytic viruses remains a significant challenge. In vitro sensitivity of malignant tumor cells to an oncolytic agent does not consistently predict therapeutic efficacy in vivo. For instance, in a study by Raquela J. Thomas et al., recombinant oncolytic myxoma virus demonstrated efficacy in two mouse models of triple-negative breast cancer, despite showing low infectivity against these cancer cells in vitro [28]. Conversely, xenografts of PDAC (BxPC-3) were resistant to VSV-FH treatment, despite the strong oncolytic activity observed in BxPC-3 cells in vitro [29]. This discrepancy may be attributed to numerous factors such as the dense desmoplastic stroma and, consequently, diminished drug delivery to PDAC tumors [30].

Intratumoral administration of immunotherapeutic agents represents an effective strategy while minimizing systemic toxicity [31], and the fact that immunization does not reduce efficacy but, on the contrary, markedly increases it [15], providing an interesting safety feature, potentially enables the use of higher doses of 17D for virotherapy. Because the direct delivery of naked viruses through intravenous injection presents significant challenges, including rapid clearance by the immune system, inadequate accumulation in tumors, and significant side effects, there is a need to develop new innovative strategies for delivering OVs [32].

The YFV vaccine strain 17D induces a strong immune responses with a mixed TH1/TH2 CD4(+) cell profile, resulting in sustained T CD8(+) responses and high titers of neutralizing antibodies [33]. Primary immunization of humans with the 17D vaccine results in early IFN-γ synthesis [34], possibly mediated by NK cells. NK cells are considered one of the most important cells secreting IFN-γ induced in the initial host immune responses in some infections, as IFN-γ is one of the most important molecules in modulating the acquired immune response. In a study on immunization of mice with vaccine virus 17D [35], it was shown that early IFN-γ production after yellow fever vaccination is also characteristic of infection in mice and is much more pronounced in C57BL/6 mice compared to BALB/c mice. C57BL/6 mice exhibited the strongest CD8(+) T-cell responses and higher titers of both neutralizing antibodies and total anti-YFV IgG. A study by Aznar et al. [12] supports the potential of the 17D vaccine for intratumoral immunotherapy and in controlling the progression of subcutaneous MC38 and B16OVA tumors in C57BL/6 mice, an effect mediated by CD8(+) T cells. In bilateral MC38 tumor-bearing mice, it was shown that CD8 T-cell depletion with an anti-CD8β mAb completely abolished 17D-induced tumor delay in those directly injected and in the contralateral tumor nodules, leading to reduced overall survival.

The positive virotherapeutic effect observed in our study with immunized animals is likely due to an additional boost of immune cells within the tumor microenvironment. Histological analysis showed infiltration of inflammatory cells (predominantly mononuclear cells) within the tumor, along with areas of necrosis and apoptosis, contrasting with the control group. In view of the fact that the virus is rapidly eliminated from the tumor and blood (as indicated by PCR results), the immune system likely contributed to the suppression of the contralateral tumor in the absence of virotherapy through the appearance of additional tumor-associated antigen presented by T cells. Therefore, the ability of 17D to penetrate Pan02 cells, slowly replicate inside, and induce virus-mediated cytostasis is crucial for the antitumor immune response. Even without lysis, the antiviral immune response to 17D can activate both innate and adaptive immunity against tumors, modulate the tumor microenvironment to reduce immunosuppression, and transform the tumor from an immunologically “cold” to “hot” state.

At this stage, we cannot conclusively state that repeated administration of 17D improves therapeutic outcomes, and it may be necessary to reconsider the intervals between virus administrations and administered dose. The observed lack of effect could be attributed to the strong immune cell infiltration within the tumor, which led to swelling and a subsequent increase in tumor size. It is possible that a continuing observation might reveal even more delayed tumor growth or complete regression; however, the mice were euthanized for humane reasons before such effects could be assessed. We do not exclude increasing resistance to the 17D due to neutralizing antibodies growth with each injection, which could have been avoided by increasing the dose of virus with each subsequent injection.

Patient-derived models, including xenografts, organoids, and explants, which are essential for studying cell communication and pancreatic cancer progression, a more accurate representation of tumor heterogeneity and complexity compared to cell-line-derived models [36]. However, the drawback of such models is the absence of an immune tumor microenvironment, which has a significant impact on the accurate assessment of the efficacy of virotherapies.

The lack of suitable animal models also poses a difficulty for investigating, specifically, YFV 17D vaccines, as immunocompetent mice are inherently resistant to YFV infection and do not support active replication of live virus, in particular, IFN-I signaling overlaps with YFV 17D replication and spread in mice. Consequently, 17D can induce only limited responses in mice, contrasting sharply with the robust immunity observed in humans [37].

An important limitation of our study is overreliance on a murine syngeneic model, whereas only in vitro experiments have been conducted with human pancreatic cancer cells. Carefully planned randomized double-blind multi-center clinical trials could define whether virotherapeutic interventions, such as YFV 17D or any other OV, provide benefits to real world patients in terms of survival or severity of the disease. This could be even more complicated since numerous aspects of interactions of novel treatments with the traditional ones are difficult to predict. Particularly, at present, chemotherapeutic regimes such as FOLFIRINOX are being used as a standard clinical recommendations. This includes a combination of at least four drugs, and every aspect of interactions with them should be tested experimentally in vitro and in vivo. CAR-T and other cell-based interventions are now intensively studied as innovative avenues, and specific immunotherapies are promising. In this situation, virotherapy may constitute a good treatment option, and 17D YFV holds great promise as an OV especially for human populations pre-immunized according to WHO recommendations—in countries with endemic wild type yellow fever virus.

## 5. Conclusions and Perspectives

In our study, we demonstrated that YFV 17D has immune-mediated potential in treating subcutaneous tumors in the syngeneic Pan02 model, with particularly enhanced effects in pre-immunized animals. Since human pancreatic cancer cell lines are more susceptible to 17D infection, it would, therefore, be interesting to investigate the oncolytic potential of YFV 17D further, for example, using in vivo tumor xenograft models or humanized mice. We hope that such studies will bring us closer to a clearer understanding of the mechanisms of tumor-specific antiviral immunity in vivo.

The live-attenuated yellow fever vaccine strain 17D has been in use for 70 years and remains the gold standard for vaccines due to its exceptional immunogenicity. It elicits a broad polyfunctional immune response, engaging multiple pathways of innate, humoral, and cellular immunity and provides long-lasting protection with a single immunization dose. Given the extensive vaccination coverage in endemic regions of Africa and Latin America, there is good potential for its success as a cancer immunotherapy.

## Figures and Tables

**Figure 1 vaccines-13-00040-f001:**
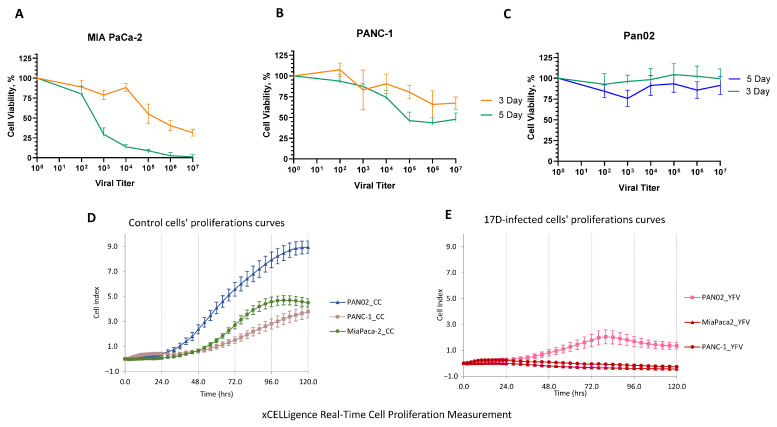
17D-induced CPE in pancreas carcinoma cell lines in vitro. (**A**–**C**) Human MIA PaCa-2, PANC-1, and mouse Pan02 pancreatic cancer cell lines were exposed to increasing titers of 17D, from 10^2^ to 10^7^ PFU/mL. Cell viability was assessed by MTT assay. The staining was performed on days three and five after exposure. (**D**,**E**) xCELLigence Real-Time cell proliferation curves of the pancreas carcinoma cells in non-infected and 17D-infected cells at MOI = 100.

**Figure 2 vaccines-13-00040-f002:**
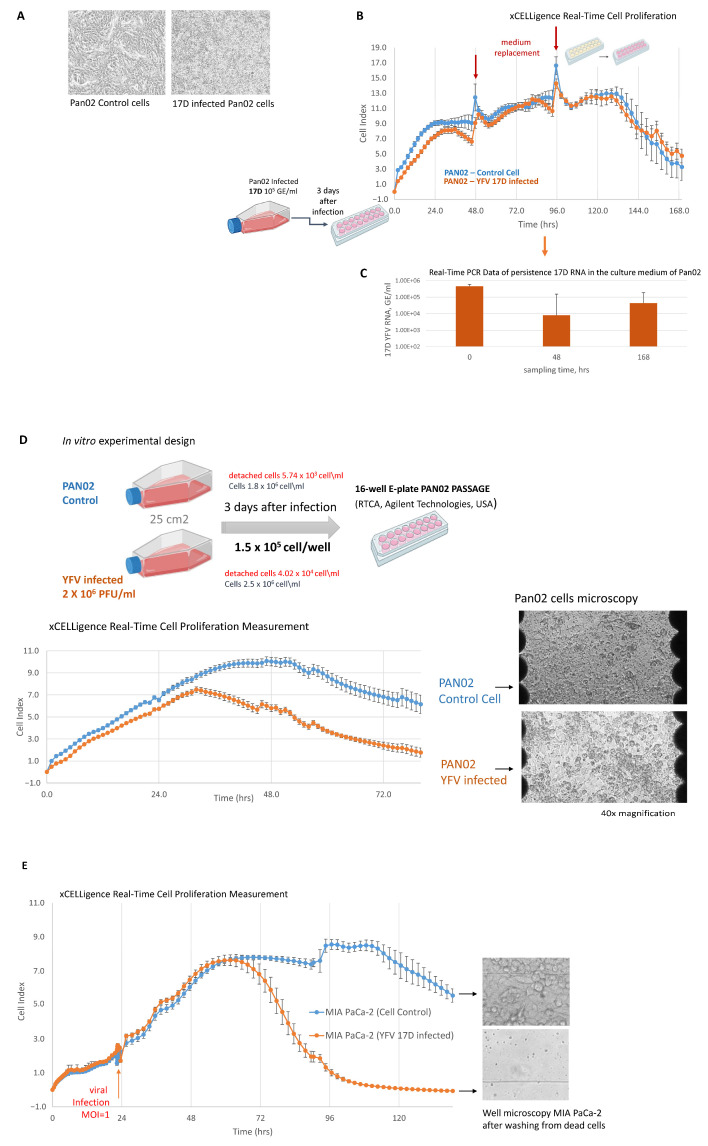
Evaluation of the CPE of 17D on the Pan02 and MIA PaCa-2 cells. (**A**) PAN02 cells at 40× magnification. The presence of detached cells in the 17D-infected Pan02 cell cultures can be observed. (**B**) Cell proliferation curves of the Pan02 cells and the 17D-infected Pan02 cells. Pan02 cells in a 25 cm^2^ cell culture flask were infected with 17D at 10^5^ GE/mL. Three days post-infection, the cells were placed on an E-plate (1.5 × 10^5^ cell/well) for xCELLigence Real-Time cell proliferation measurement. The medium was replaced after 48 and 96 h. (**C**) Real-Time PCR data of the persistence of 17D RNA in the culture medium of Pan02 cells. (**D**) 17D required a longer time to exert a CPE in the mouse cell line Pan02 than in the human cells. Pan02 cells in a 25 cm2 cell culture flask were infected with 10^6^ PFU/mL of 17D and incubated for three days. The detached cells were counted on the third day, following the passage of Pan02 cells on an E-plate for the cell proliferation measurement. The proliferation curves show delayed growth of the 17D-infected Pan02 cells. Microscopy of the Pan02 cells, at 40× magnification, showed an increase in the number of detached cells. (**E**) 17D-induced a CPE in human MIA PaCa-2 cells by xCELLigence Real-Time cell proliferation measurement. The infected Mia PaCa2 cell proliferation curve shows that after 60 h, the infected cells began to detach and were subsequently lysed. Microscopy on cells at 100× magnification.

**Figure 3 vaccines-13-00040-f003:**
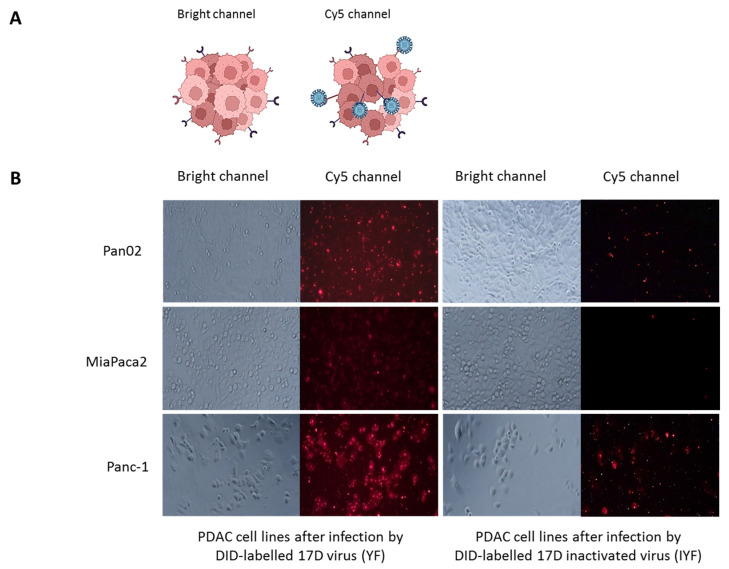
17D infection of Pan02 cells. (**A**) Scheme of the experiment. The DID-labeled virus’s binding with a cell membrane emits a fluorescent signal visible in the Cy5 channel after the DiD molecules, embedded in the viral membrane, move away from each other. (**B**) Analysis of the fusion of cellular and viral membranes for Pan02, MIA PaCa-2, and PANC-1 cells after infection by the DID-labeled infectious 17D and the DID-labeled inactivated 17D. Cells’ microscopy at 200× magnification.

**Figure 4 vaccines-13-00040-f004:**
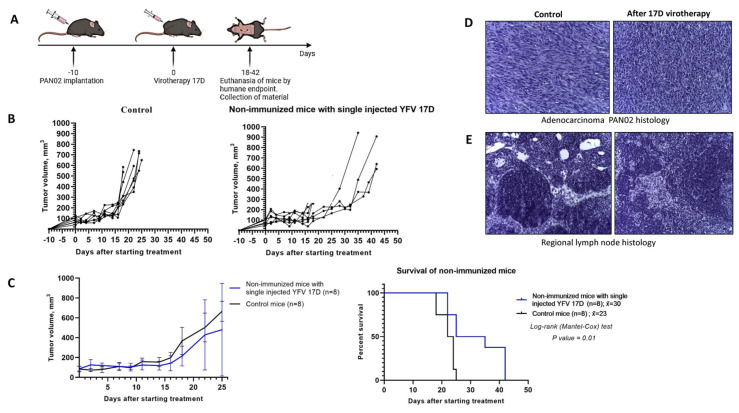
Intratumoral administration of 17D inhibits Pan02 tumor growth and improves survival time in a murine model of pancreatic cancer. (**A**) Schematically depicted experimental phases: Pan02 tumors engrafting and treatment with YFV 17D. Specifically, 10^6^ PFU/100 μL of 17D was administered intratumorally for the virotherapy group, and 100 μL DMEM culture medium from uninfected Vero cells was administered intratumorally for the control group. Upon reaching a humane endpoint, defined as a tumor greater than 10 mm in two dimensions or a significant decline in health, potentially causing suffering (limb atrophy, starvation, and critical weight loss), mice were euthanized, necropsied, and their blood and tumor samples collected. (**B**) The graphs represent individual tumor size after intratumoral injections of 17D (n = 8) in the virotherapy group and DMEM (n = 8) in the control group. (**C**) Overall survival graphs are shown as the mean ± SD and Kaplan–Meier survival analysis. Long-rank (Mantel–Cox) test, *p* = 0.01. (**D**) Histological study of tumors in the control group and the group with a single intratumoral virotherapy with 17D. Pan02 adenocarcinoma fragment at day 25 from control mouse (left) and 17D virotherapy mouse (right). Stained with hematoxylin and eosin; 200× magnification. Peritumoral zone of the tumor with diffuse mononuclear infiltration. Tumor of partially solid structure with active proliferation of atypical cells, necrosis and diffusely located lymphocytes. (**E**) A fragment of a regional lymph node in Pan02-bearing mice from the control group and the group with a single intratumoral virotherapy euthanized at days 25 (left) and 35 (right). Stained with hematoxylin and eosin; 200× magnification. Regional lymph nodes are not enlarged, lymphoid follicles are sparse, germinative centers are small, metastases were not found.

**Figure 5 vaccines-13-00040-f005:**
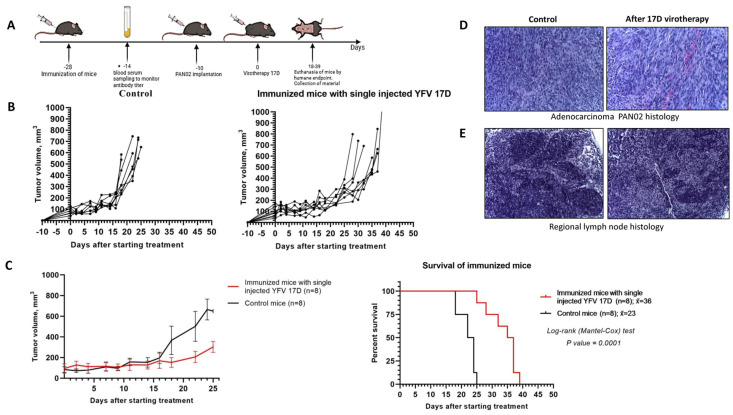
17D pre-immunization enhances the efficacy of the intratumoral 17D-therapy. (**A**) Schematically depicted experimental phases of immunization with 17D, Pan02 tumors engrafting, and the treatment with YFV 17D. C57BL/6 mice were pre-immunized with 10^6^ PFU/100 μL of 17D by subcutaneous injection at the base of the tail; the sera from the 17D-immunized mice were collected at 14 day post-immunization for neutralization assay. Then, after 4 days, 10^6^ PFU/100 μL of 17D was administered intratumorally for the virotherapy group, and 100 μL DMEM culture medium from uninfected Vero cells was administered intratumorally for the control group. Upon reaching a humane endpoint—defined as a tumor greater than 10 mm in two dimensions or a significant decline in health, potentially causing suffering (limb atrophy, starvation, critical weight loss)—mice were euthanized, necropsied, and their blood and tumor samples were collected. (**B**) The graphs represent individual tumor sizes of the immunized mice after intratumoral injections of 17D (n = 8) or DMEM (n = 8) as a control group. (**C**) Overall graphs are shown as the mean ± SD and Kaplan–Meier survival analysis. Long-rank (Mantel–Cox) test with *p* = 0.0001. (**D**) A histological examination of tumors from immunized mice of the control group and the group treated with a single intratumoral injection of 17D. The Pan02 adenocarcinoma fragment at day 25 of a mouse from the control group (left); the Pan02 adenocarcinoma fragment at day 35 from a mouse from the 17D virotherapy group (right), stained with hematoxylin and eosin, 200× magnification. Peritumoral zone with moderate inflammatory cell infiltration. The tumor consists predominantly of a trabecular–fascicular structure and contains large foci of necrosis and apoptosis of atypical cells, as well as a weak diffuse mononuclear infiltration. (**E**) A fragment of a regional lymph node in Pan02-bearing immunized mice from the control group and the group with a single intratumoral injection of 17D at days 25 (left) and 35 (right), stained with hematoxylin and eosin, 200× magnification. Regional lymph nodes are not enlarged, lymphoid follicles are sparse, germinative centers are small, and metastases were not found.

**Figure 6 vaccines-13-00040-f006:**
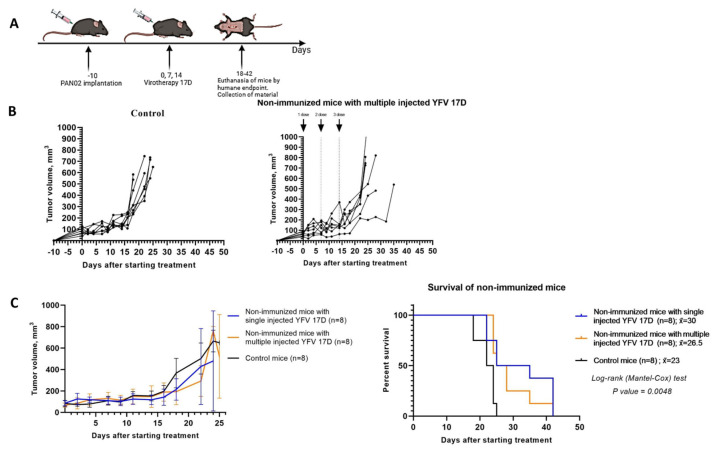
Multiple intratumoral administration of 17D in non-immunized mice. (**A**) Schematically experimental phases: Pan02 tumor engrafting and YFV 17D treating representation. 5.2 × 10^5^ Pan02 cells in 100 μL PBS were implanted in the right flank C57BL/6 mice, 10 days after the tumor reached a size of 5–7 mm, 17D was administered intratumorally at a dose 10^6^ PFU/100 μL (treatment group) or 100 μL DMEM culture medium from Vero cells (control group) triple at days 0, 7, and 14. Mice were euthanized upon reaching humane endpoint (tumor greater than 10 mm in two dimensions or critical health condition), autopsied, and blood and tumor samples were collected. (**B**) Graphs represent individual tumor sizes in mice after multiple intratumoral injections of 17D (n = 8) or medium (n = 8) as control group. (**C**) Tumor growth graphs shown as mean ± SD and Kaplan–Meier survival analysis in mice after single intratumoral injections of 17D (n = 8), multiple injections of 17D (n = 8), or media (n = 8) as a control group. Long-rank (Mantel–Cox) test; *p* < 0.0048.

**Figure 7 vaccines-13-00040-f007:**
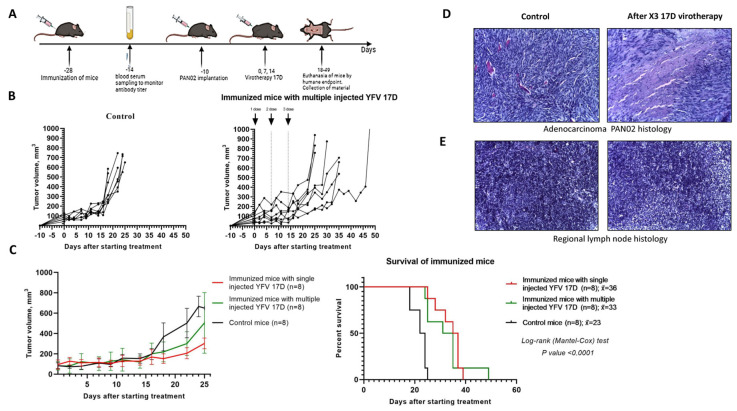
Multiple 17D intratumoral administration in group of immunized mice. (**A**) Outline of experimental phases: 17D pre-immunization, Pan02 tumor engrafting, and YFV 17D treatment. C57BL/6 mice were preliminary immunized with 10^6^ PFU/100 μL 17D by subcutaneous injection at the base of the tail, sera from 17D-immunized mice were collected at 14-day post-immunization for a neutralization assay. Then, after four days, 5.2 × 10^5^ Pan02 cells in 100 μL PBS were implanted in the right flank C57BL/6 mice, day 10 after the tumor reached a size 5–7 mm, 17D was administered intratumorally at a dose 10^6^ PFU/100 μL (for treatment group) or 100 μL DMEM culture medium from VERO cells (for Control group) with repeated injection at day 7 and 14. Mice were euthanized upon reaching humane endpoint, autopsied, and blood and tumor samples were collected. (**B**) Graphs represent individual tumor size, immunized mice after multiple intratumoral injections of 17D (n = 8) or media (n = 8) as a control group. (**C**) Tumor growth graphs shown as the mean ± SD and Kaplan–Meier survival analysis immunized mice after intratumoral single injections 17D (n = 8), immunized mice after multiple intratumoral injections of 17D (n = 8) or media (n = 8) as a control group. Long-rank (Mantel–Cox) test *p* < 0.0001. (**D**) Histological examination of tumors from immunized mice of the control group and multiple (×3) intratumoral 17D virotherapy group. Pan02 adenocarcinoma at day 25 from control mouse (left) and day 35 after 3x17D virotherapy mouse (right). Stained with hematoxylin and eosin 200× magnification. For the therapeutic group, the paratumoral zone of the tumor with moderate inflammatory cell infiltration. The tumor is predominantly of trabecular-fascicular structure, contains medium foci of necrosis and apoptosis of atypical cells, as well as diffuse weak mononuclear infiltration. (**E**) A fragment of a regional lymph node in Pan02-bearing immunized mice from the control group and single intratumoral virotherapy group at day 25 (left) and day 35 (right). Stained with hematoxylin and eosin, 200× magnification. Regional lymph nodes are not enlarged, lymphoid follicles are sparse, germinative centers are small, no metastases were found.

**Figure 8 vaccines-13-00040-f008:**
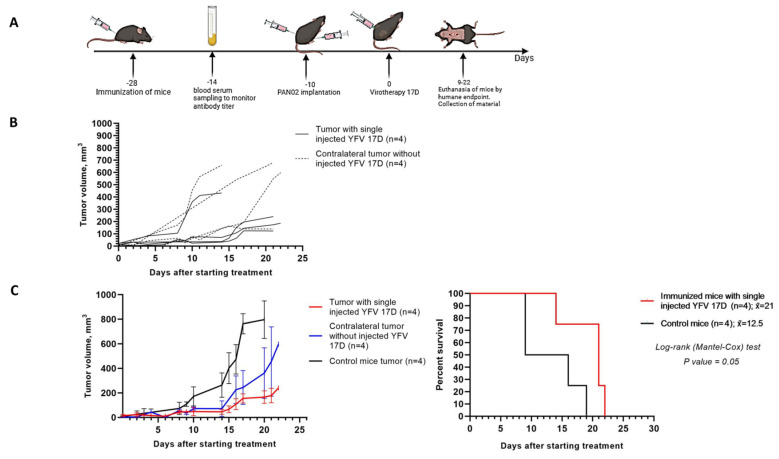
Immunized mice exhibit delayed contralateral tumor growth. (**A**) Schematically depicted experimental phases of immunization with 17D, Pan02 tumors engrafting, and YFV 17D treating representation. C57BL/6 mice were preliminarily immunized with 10^6^ PFU/100 μL of 17D by subcutaneous injection at the base of the tail, and sera from 17D-immunized mice were collected at day 14 post-immunization for the neutralization assay. After four days, 5.7 × 10^5^ Pan02 cells in 100 μL PBS were implanted in the both flanks of C57BL/6 mice; at day 10 after the tumor reached a size of 2.5–3.5 mm, 17D was administered intratumorally at a dose of 10^6^ PFU/100 μL (treatment group) or 100 μL DMEM culture medium from Vero cells (control group) into flanks. Mice were euthanized upon reaching the humane endpoint. (**B**) Graphs represent individual tumor size of immunized mice after intratumoral injections of 17D (n = 4) or contralateral tumor without injection of 17D (n = 4). (**C**) Tumor growth graphs shown as the mean ± SD and Kaplan–Meier survival analysis of immunized mice after intratumoral single injections of 17D (n = 4) or media (n = 4) as a control group. Long-rank (Mantel–Cox) test *p* = 0.05.

**Figure 9 vaccines-13-00040-f009:**
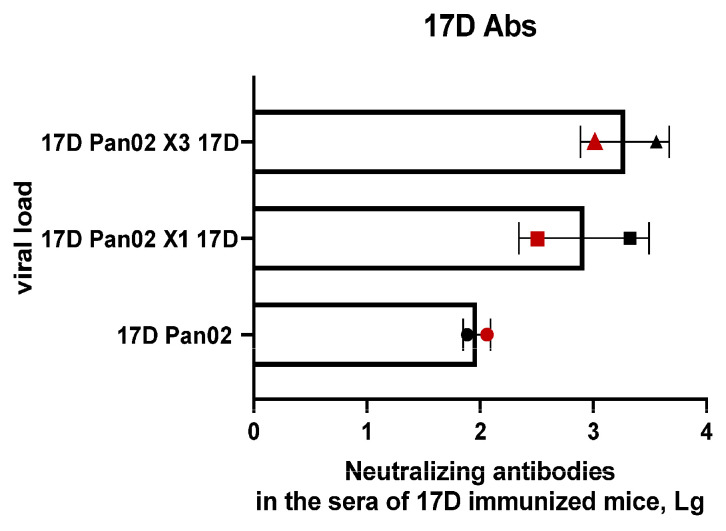
Viral load and neutralizing antibodies during therapy. Neutralizing antibodies in the serum of 17D-immunized mice (n = 8/group). 17D Pan02: control group—17D-immunized mice with tumors, without 17D virotherapy; 17D Pan02 X1 17D: 17D-immunized mice with tumors and single-dose 17D virotherapy; 17D Pan02 X3 17D: 17D-immunized mice with tumors and triple-dose 17D virotherapy. Symbols represent different serum collection time points: red—day 14 after immunization; black—humane endpoint.

**Figure 10 vaccines-13-00040-f010:**
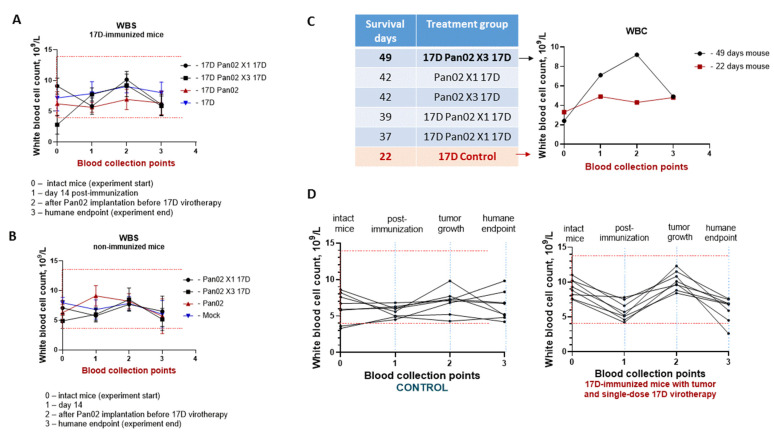
White blood cell (WBC) count in 17D-immunized C57BL/6 mice with Pan02 tumors: (**A**,**B**) presented as the median for each group—(**A**) immunized mice groups; (B) non-immunized mice groups. WBC count for C57BL/6 mice at various time points: 0—intact mice (experiment start); 1—day 14 post-immunization; 2—after Pan02 implantation, before 17D virotherapy; 3—humane endpoint (experiment end). Red dashed lines—reference values for the C57BL/6 mice. (**C**) WBS parameters for the individual mice: one responded most effectively to virotherapy (49 survival days) and the control mouse with the most aggressively growing tumor (22 survival days). (**D**) WBS parameters for the immunized mice group with virotherapy and the control.

**Table 1 vaccines-13-00040-t001:** Real-time PCR data on the persistence of 17D RNA in the culture medium and Pan02 cells, genome equivalents (GE/mL).

Sample	Days After Infection			
	3	7	10	13
Culture medium, GE/mL	6.8 × 10^4^	2.9 × 10^5^	6.1 × 10^4^	4.7 × 10^4^
Cells, GE/mL	NI	NI	4.0 × 10^5^	1.7 × 10^5^

**Table 2 vaccines-13-00040-t002:** Real-time RT-PCR and PFU assay data on the persistence of 17D RNA during the culture of Pan02 cells.

Virus Persistance	Infection Day MOI	3 Days After Infection	1st Passage	2nd Passage	3rd Passage
PFU/mL	2.5 × 10^6^	3 × 10^3^	3 × 10^3^	1 × 10^3^	3 × 10^2^
PCR, GE/mL	1 × 10^6^	1.3 × 10^6^	4.5 × 10^5^	2.3 × 10^5^	1.2 × 10^5^

**Table 3 vaccines-13-00040-t003:** RT-PCR assay data on the persistence of 17D RNA in the culture medium of human pancreatic carcinoma cells.

Virus		17D YFV		
Cell Lines	Days	0	3	5
MIA Paca-2	PCR gE/mL	1 × 10^6^	2.3 × 10^8^	1.6 × 10^8^
PANC-1	PCR gE/mL	1 × 10^6^	9.6 × 10^7^	6.2 × 10^7^

**Table 4 vaccines-13-00040-t004:** Neutralizing 17D antibodies in the sera of the 17D-immunized mice after day 14 post-immunization and at the end of the experiment.

Group	Mouse No.	Blood Collection After Immunization Neutralizing Antibodies, Lg	Blood Sampling at the End of the Experiment Neutralizing Antibodies, Lg
17D-immunized mice with tumor and single-dose 17D virotherapy	1	2.8	3.4
	2	2.4	2.8
	3	2.9	>3.4
	4	2.4	>3.4
	5	2.4	>3.4
	6	2.4	3.4
	7	2.4	3.4
	8	2.4	3.4
17D-immunized mice with a tumor and triple-dose 17D virotherapy	1	3	>3.4
	2	3	>3.4
	3	2	3.1
	4	3.3	3.4
	5	3.4	>3.4
	6	2.9	>3.9
	7	3.4	>4.3
17D-immunized mice with a tumor, without 17D virotherapy (Control)	1	2	1.5
	2	2	<2
	3	2	1.7
	4	2.1	<2
	5	2.1	2
	6	2.1	2
	7	2.1	2

C57BL/6 mice were immunized with 10^6^ PFU/100 μL of 17D by subcutaneous injection at the base of the tail. Samples of the sera from the 17D-immunized mice were collected at day 14 for the neutralization assay.

## Data Availability

The data presented in this study are available in this article.

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
