# Peer review of "Immunotherapeutic Potential of the Yellow Fever Virus Vaccine Strain 17D for Intratumoral Therapy in a Murine Model of Pancreatic Cancer"

_vaccines, 2025, doi:10.3390/vaccines13010040_

Round 1

Reviewer 1 Report (Previous Reviewer 1)

Comments and Suggestions for Authors

This study has been significantly improved and it can be accepted after a minor revision. 

Comments on the Quality of English Language

This manuscript is a resubmission of the previous manuscript #3280834.

Even though the authors did not list a point-by point responses to the reviewers, I looked up my previous review and then evaluate this newer version of the manuscript and found that the authors have responded to at least some of my comments.

Overall, with some additional data now presented, this study is more complete, and data are stronger and solid to support the authors’ conclusions.

There are two minor issues:

1. there are a few typos and some minor grammatic errors in the manuscript. For example, (1). Line 332. The fused word ‘Mediumand’ should be separated into “medium and…”. (2). Line 337. ‘mediumPan02” should be ? (3). Line 347. “in the culture medium human pancreas carcinoma, where it should add the word ‘of’ between ‘medium’ and ‘human’.  (4). Line 448. The section subtitle “17D immunization…” could be replaced by “17D preimmunization…” to better reflect the real situation.

2. In ref #18, the name of the journal is missing.

Author Response

We thank you for your careful consideration of our manuscript. In the first version we were attentive to every even the smallest remark and only because of this the manuscript has improved significantly. 
Apparently there was a technical error, because we provided a response to each comment in letters to the reviewers and uploaded everything to the MDPI submission system, maybe we forgot to upload a separate file.

Comments 1 - 1. there are a few typos and some minor grammatic errors in the manuscript. For example, (1). Line 332. The fused word ‘Mediumand’ should be separated into “medium and…”. (2). Line 337. ‘mediumPan02” should be ? (3). Line 347. “in the culture medium human pancreas carcinoma, where it should add the word ‘of’ between ‘medium’ and ‘human’.  (4). Line 448. The section subtitle “17D immunization…” could be replaced by “17D preimmunization…” to better reflect the real situation.

Response 1 - All typos have been corrected (yellow marking).

Comments 2 In ref #18, the name of the journal is missing. 

Response 2 - In ref #18, the name of the journal was inserted (yellow marking).

Reviewer 2 Report (Previous Reviewer 3)

Comments and Suggestions for Authors

The revised manuscript is indeed an improvement. However, it would have enormously helped the reviewing if the authors had bothered to highlight ALL changes made from the original version. Please make the final revision as suggested below.

General Comments

Although the authors have revised the manuscript to some extent, the following points should be considered for additional revision.

Please revise the sentence (L36) “In light of the fact that YFV 17D in vitro affected human cancer cells much stronger than the mice's, these findings appear promising” to “The fact that YFV 17D  affected human cancer cells much stronger than mice cancer cells in vitro appears promising. Still, the authors do not present any reasons for it being the case in vivo. On the contrary, many studies providing good results in vitro have proven disappointing in animal models and even worse in human trials.

Specific Comments

L117: : “in the work” > “in the study” 

Comments on the Quality of English Language

Please go through the text once more!

Author Response

Thank you for these comments. We tried to correct all typos, and improve English, especially by clarification of the ambigous sentences. Imprortant additions are marked in color, and a text comparison file is also included separately.

Reviewer 3 Report (New Reviewer)

Comments and Suggestions for Authors

The cytopathic impact of YFV 17D on both human-derived and murine syngeneic pancreatic cancer cells was rigorously evaluated via in vitro and in vivo experimentation. The results revealed that YFV 17D significantly impaired the viability of human-derived cancer cells in vitro. While no lytic effect was observed on Pan02 murine cells in vitro, a single intratumoral administration of 17D notably retarded tumor progression and enhanced median survival rates by 30%. Additional injections of 17D did not further inhibit tumor growth but considerably extended the median survival duration. Moreover, priming with 17D immunization amplified its therapeutic potency. Given the markedly pronounced effect of YFV 17D on human-derived cancer cells compared to murine cells in vitro, these results are deemed encouraging. Consequently, it is anticipated that the therapeutic efficacy of YFV 17D-based oncolytic treatment for human pancreatic cancer in vivo may surpass that observed in murine pancreatic tumors.

In my judgment, this manuscript demonstrates significant merit, exhibiting notable innovation and research value. The scope of work satisfies the requirements. Following minor revisions, it is recommended for acceptance. The specific revision suggestions are outlined below.

1. To address the constraints of relying solely on the mouse model, the authors might consider incorporating pertinent clinical studies. Such inclusions could offer a more comprehensive assessment of the efficacy and safety of YFV 17D in treating human pancreatic cancer, thereby bolstering the evidence for its clinical utility.

2. It is advised that a comparative analysis be conducted between YFV 17D and other prevalent treatments for pancreatic cancer, including chemotherapy, radiotherapy, and immunotherapy. This comparison will elucidate the therapeutic benefits and the specific applications of YFV 17D, thereby enabling the formulation of personalized treatment plans for patients.

3. Further investigation into the behavior of YFV 17D in mice is recommended, specifically focusing on its persistence, immunogenicity, and tissue distribution. This will elucidate the mechanism of action of YFV 17D in treatment and potentially stimulate innovative approaches to enhance treatment methodologies. Concurrently, it is imperative to assess whether prolonged usage of YFV 17D could induce adverse reactions or immune suppression.

4. It is recommended to further improve the resolution of the figure.

5. It is recommended to cite the following literature:

[1] A. Gu, J. Li, S. Qiu, S. Hao, Z.-Y. Yue, S. Zhai, M.-Y. Li, Y. Liu, Pancreatic cancer environment: From patient-derived models to single-cell omics. Mol. Omics 2024, 20, 220233. DOI: 10.1039/D3MO00250K

[2] K. Xiao, Y. Lai, W. Yuan, S. Li, X. Liu, Z. Xiao, H. Xiao, mRNA-based chimeric antigen receptor T cell therapy: Basic principles, recent advances and future directions. Interdisciplinary Medicine 2024, 2, e20230036. https://doi.org/10.1002/INMD.20230036

[3] Wang J, Liao Z-X. Research progress of microrobots in tumor drug delivery. Food & Medicine Homology, 2024, 1(2): 9420025. https://doi.org/10.26599/FMH.2024.9420025

Author Response

This manuscript is a resubmission of an earlier submission. The following is a list of the peer review reports and author responses from that submission.

Round 1

Reviewer 1 Report

Comments and Suggestions for Authors

The authors have evaluated the oncolytic potential of yellow fever virus vaccine strain 17D (YFV 17D) in a few pancreatic cancer cell lines of both murine and human origins, and then evaluated its immunotherapeutic potential on murine pancreatic Panc02 tumor model.

The study is interesting and worth publishing after revisions to address some issues satisfactorily.

There are a few questions.

1.      The viral dose used. The authors have used viral dose of 1.0 x 10(6) PFU/100 uL in in vivo experiments to achieve minimal therapeutic effect, albeit statistically significant. Why didn’t the authors use a higher dose? Is it due to technical issues?

2.      Section 3.5. The study using pre-immunized mice. It is not clear how this works. Do authors have some good scientific explanations?

3.      The manuscript lacks a real section 3.3. Thus, please re-label sections 3.4 to 3.6

4.      The authors tested its cytotoxicity of the virus on panc02 as well as human pancreatic cancer cells in vitro and concluded that panc02 cells are not so susceptible to this virus when compared to the human pancreatic cancer cells. This may change once the panc02 cells are implanted to syngeneic B6 mice as the in vivo conditions and tumor microenvironment modulate gene expression in host cancer and stromal cells and potentially susceptibility of the cancer cells to a virus infection. Have the authors examined this possibility?

5.      Page 7 of 18, lines 323-324. The phrase “17D was administered intratumorally at a dose of…” is redundant. Please delete it.

6.      Figure 2A. It is not clear what the 2 pictures on the right side represent. What is the difference between them?

7.      Figure 5D. To draw solid conclusions, the authors need to add a control group, tumors from mice without virotherapy.

8.      It needs some grammatic checking throughout the manuscript. For example, line 355. “preliminary immunized”.

9.      There are wrong formats or minor errors/misses in the following references:

Ref #3; 4; 7; 9 (missing article number); 10; 19; 20; 21; 22; 27; 28; 29; 31; 34; 35.

Author Response

Dear Colleague! On behalf of the team of authors we express our great pleasure afor your attention to our work.

Comment 1.      The viral dose used. The authors have used viral dose of 1.0 x 10(6) PFU/100 uL in in vivo experiments to achieve minimal therapeutic effect, albeit statistically significant. Why didn’t the authors use a higher dose? Is it due to technical issues?

Response1: In the design of the experiment, we relied on the fact that the virus titer in the human yellow fever vaccine is even lower to lg5 and that this would be a highly significant viral load for the mouse. We purified the virus, but purification is indeed accompanied by technical difficulties and significant losses of its activity. Therefore, in the design of the experiment a triple sequential inoculation group was introduced to increase the expected effect, although this group did not meet our expectations. Also in the design of the experiment we relied on previous studies, e.g. Aznar et al in which reported efficient results tumor therapy at the same doses of lg6 17D.

Comment 2.      Section 3.5. The study using pre-immunized mice. It is not clear how this works. Do authors have some good scientific explanations?

Response2: We agree with your comment absolutely. This term is generally misleading to the reader. Although we have also adopted it from other works. We have revised the text and rewritten it to simply immunized mice and non-immunized mice.

Comment 4.  The authors tested its cytotoxicity of the virus on panc02 as well as human pancreatic cancer cells in vitro and concluded that panc02 cells are not so susceptible to this virus when compared to the human pancreatic cancer cells. This may change once the panc02 cells are implanted to syngeneic B6 mice as the in vivo conditions and tumor microenvironment modulate gene expression in host cancer and stromal cells and potentially susceptibility of the cancer cells to a virus infection. Have the authors examined this possibility?

Response 4: Yes, they did, and given the known fact that in principle mouse 17D is sensitive to human interferon, they realized that the syngeneic model is very weak. In conclusions pointed this out:

"In our study, we demonstrated that YFV 17D has immune-mediated potential in treating subcutaneous tumors in the syngeneic Pan02 model, with particularly enhanced effects in immunized animals. Since human pancreatic cancer cell lines are more susceptible to 17D infection and it is therefore interesting to investigate the oncolytic potential of YFV 17D further, for example, using in vivo tumor xenograft models, and even better, using humanized mice overlapping the disadvantage of this model due to susceptibility 17D to murine IFN-I. We expect that overlaying the two models of mice with human tumor xenografts and humanized mice with overlapping deficiencies will bring us closer to a clearer prediction of the mechanisms of tumor-specific antiviral immunity in vivo."

Response 3, 5-9 comments: Fully agree. The remarks are  formal. All corrections have been made to the text. Thanks.

We provide a new version of the paper with additions and new data, taking into account all comments from all reviews. Figures at the end.

Reviewer 2 Report

Comments and Suggestions for Authors

Review Vaccines 10/24

The manuscript titled “Immunotherapeutic potential of the yellow fever virus vaccine strain 17D for intratumoral therapy in a murine model of pancreatic cancer” describes the effects of virus vaccine 2 strain 17D regarding cell proliferation, cytopathic effects and fusion of the yellow fever virus vaccine strain 17D with pancreatic cancer cells (both murine PAN02 and human PANC-1, MIA PaCa-2) in vitro. Furthermore, data are presented from a syngeneic PAN02 tumor model in C57BL/6 mice. The data presented are somehow in line with data from the literature demonstrating similar effects of yellow fever virus vaccine in other murine models. The manuscript is well written, and the data is presented clearly. Overall, the effect seem to be moderate in the murine model and only account for the survival data and not for the tumor growth data.

A major weakness of the data set lies in the fact, that it is only descriptive and that no mechanistic insights are provided regarding the immunological background of the findings. This however appears to be essential, considering that in vitro cell proliferation was not influenced in PAN02 cells in vitro.

The finding that “pre-immunization” (the term preliminary 17D immunization seems inappropriate) with the yellow fever virus vaccine strain 17D enhanced the effect of intra-tumoral injection of the vaccine regarding tumor growth and survival is very interesting, however, it would be essential to gain some more information on the role of cellular immune responses in this respect. 

Moreover, a second tumor model using mice with a different genetic background could give an answer to the question if the “pre-immunization” is also enhancing the effects under these conditions.

Author Response

Dear Colleague! On behalf of the team of authors we express our great pleasure for your attention to our work.

Comment 1: A major weakness of the data set lies in the fact, that it is only descriptive and that no mechanistic insights are provided regarding the immunological background of the findings. This however appears to be essential, considering that in vitro cell proliferation was not influenced in PAN02 cells in vitro.

Response 1: Completely agree with your assertion! We have substantially revised the narrative of the article and refined the logicality of the presentation. We've added two sections:

3.7. Delayed contralateral tumor growth in the immunized mice

3.8. Viral load, neutralizing antibodies, and white blood cells analysis

We've expanded the evidence base for immunotherapy a bit by revisiting some of the data.

Unfortunately, the evaluation of the contribution of a specific T-lymphocyte subpopulation for 17D-therapy was not performed in this study, although this was studied in mice models in the previous works.

However, we now realize that this syngeneic model is not the most suitable for investigating 17D-therapies and see no point in further extending studies on it. Moreover, our colleagues Aznar et al have already done all this with C57bl and there is no point in repeating it.

But in perspective since human pancreatic cancer cell lines are more susceptible to 17D infection and it is therefore interesting to investigate the oncolytic potential of YFV 17D further, for example, using in vivo tumor xenograft models, and even better, using humanized mice overlapping the disadvantage of this model due to susceptibility 17D to murine IFN-I. We expect that overlaying the two models of mice with human tumor xenografts and humanized mice with overlapping deficiencies will bring us closer to a clearer prediction of the mechanisms of tumor-specific antiviral immunity in vivo.

Comments 2 pre-immunization” (the term preliminary 17D immunization seems inappropriate) 

Response 2: We agree with your comment absolutely. This term is generally misleading to the reader. Although we have also adopted it from other works. We have revised the text and rewritten it to simply immunized mice and non-immunized mice.

Comments 3: Moreover, a second tumor model using mice with a different genetic background could give an answer to the question if the “pre-immunization” is also enhancing the effects under these conditions.

Response 3: we plan to continue using in vivo tumor xenograft models, and even better, using humanized mice overlapping the disadvantage of this model due to susceptibility 17D to murine IFN-I. We expect that overlaying the two models of mice with human tumor xenografts and humanized mice with overlapping deficiencies will bring us closer to a clearer prediction of the mechanisms of tumor-specific antiviral immunity in vivo.

We provide a new version of the paper with additions and new data, taking into account all comments from all reviews. Figures at the end.

Sincerely Y. Biryukova et al

Reviewer 3 Report

Comments and Suggestions for Authors

The manuscript titled “Immunotherapeutic potential of the yellow fever virus vaccine 2 strain 17D for intratumoral therapy in a murine model of pan-3 creatic cancer” describes the application of the YFV 17D strain vaccine for evaluation for immune responses in human and mouse pancreatic tumor cell lines and intratumoral administration in mice with implanted pancreatic tumors. The study demonstrated strong cytopathic effects in vitro and delayed tumor growth and a moderate extension of median survival in mice.

The study is generally relatively well-planned and executed. However, the writing is of poor quality and needs some major revisions not only to the language but also to the scientific presentation before it can be approved for publication. It is highly recommended that the points raised below should be taken into account.

General comments

The authors claim (L28-31) that as the YFV D17 vaccine showed a superior cytopathic effect on human cells compared to mouse cells the oncolytic therapy is anticipated to be better against human pancreatic cancers than mouse equivalents in vivo.    

“Oncolytic viruses” (L55) appear in the text without any further description or references. It should be stated that oncolytic viruses specifically replicate in tumor cells killing them while the normal cells are generally unaffected.

What are “experimental mice” (L255)?

What is the statement “the therapeutic impact in in vivo experiments could be even more pronounced” (L490) based on? Compare this statement to “In vitro sensitivity of malignant tumor cells to an oncolytic agent does not consistently predict therapeutic efficacy in vivo” (L496), which contradicts the previous one!

The authors discuss intratumoral administration as favorable to systemic delivery based on toxicity (L504) without even mentioning that typically oncolytic viruses replicate specifically in tumor cells and cause no or relatively little harm to normal tissue.

I am not sure the authors have understood the mechanism of Sindbis virus (a self-replicating RNA virus belonging to alphaviruses) infections (L537) as in the references [32, 33] replication-deficient vectors are used triggering bystander immunity.

The authors claim that the lack of superior therapeutic effect after repeated administration is due to “The observed lack of effect could be attributed to strong immune cell infiltration within the tumor” (L549) instead of simply considering the possible resistance against YFV after the first immunization.

The statement “The live-attenuated yellow fever vaccine strain 17D has been in use for over ninety years” (L569) is incorrect. Although the YFV was discovered approximately 90 years ago, the 17D strain has been used for 70 years.

Specific comments

L22: “studies” > “studied”

L40: “in KRAS” > “in the KRAS”

L44: “operable patients” does not make sense as “chemoradiotherapy” has nothing to do with surgery

L53: There is no point in using capital letters for the different viruses, except for “Newcastle disease virus”, which is missing from the last. Use also “viruses” instead of “virus” as different types are applied.

L61: “their” > “the”

L71: “virus concentrations that are innocuous for non-transformed human cells” is not clear, please revise.

L86: “In this work” > “In this study”

L94: “epithelial cells....was” > “epithelial cells....were”

L97: “from collection” > “from the collection”

L105: “were negative” > “were tested negative”

L109: “To create” > “To produce”

L110: “infection dose”???

L111: “3rd” > “third”

L128: “A sample of the culture medium was taken in a volume of 1 ml” > “A 1 ml sample of the culture medium was used”

L137: “Quantitative” > “quantitative”

L153: “For recording proliferation curves” > “For determination of proliferation curves”

L155: “recorded” > “measured”

L158: “virus cytopathic effect” > “the cytopathic effect of the virus”

L161: “applied already infected cells in growth medium (1.5 x 105 cells/well) or applied cells....” is not clear.

L183: “weighing 6-7 weeks” does not make sense as weight is measured in grams not time!

L185: “procedures was” > “procedures were”

L196: “groups without virotherapy (Control)” > “control group”

L197 & elsewhere: “VERO” > “Vero”

L213: Delete “without virotherapy”.

L231: “CO2” > “CO2

L244: “number” > “the number”

L259: “17D virus cytopathic effects on pancreatic carcinoma cell lines in vitro on” > “Cytopathics effects of the 17D virus on pancreatic carcinoma cell lines in vitro”

L264: “5 days from exposure” > “five days of exposure”

L271: “multiplicity of infection” > “multiplicity of infection (MOI)”

L271: “MOI 100/cell” > “MOI 100”

L273: Delete “is”

L276: “Pan02 cell layer” > “Pan02 cells”

L281: “To give a putative mechanism” is poor language!

L285 x2: “cells\ml” > cells/ml”

L291: “cytopathic effect (CPE)”; the abbreviation CPE should be mentioned earlier in thetx.

L294 “hours” is used, while on L296 & 297 it is “h”; be consistent!

L295, 296 & 298: What does “gE/ml” stand for?

L300: “3” > “three”

L311: “into the cells” > “into cells”

L322 & elsewhere: “right flask” > “right flank”

L323: The repetition in “was administered intratumorally at a dose of, 17D was administered intratumorally at a dose 106 PFU/100 µl” does not make sense.

L332: “easily tolerated” > “well tolerated”

L347: “at 14-day” > “at day 14”

L378: Delete “does”

L379: “10^6” > “106

L402: “7” > “seven”

L410: “Cytopathic Effects” > “CPEs”

L412: “by MTT test” > “by the MTT test”

L413: “3 and 5” > “three and five”

L417: “PAN02 cells layer and cells after 3” > “PAN02 cells layer and cells

L418: “cells has no......and is” > “cells have no......and are”

L422: “105 cell/well” > “105 cells/well”

L429 & elsewhere: “Magnification x200” > “200x magnification”

L443 & elsewhere: “25th...35th day” > “day 25 and 35”

L469: “14-day” > “day 14”; “after 4” > “after four”

L470: “10 days” > “ten days”

L474: “mouse's malaise”, please revise!

L508: “yellow fever virus” > “YFV”

L525: The sentence “contrasting with both the control group and non-immunized mice without virotherapy” does not make sense. What is the difference between the control group and the non-immunized mice? It is also obvious that non-immunized mice have not received virotherapy!

L531: “lysys” > “lysis”

L544: “vaacine” > “vaccine”

Comments on the Quality of English Language

The language is poor and requires a major revision.

Author Response

Dear Colleague! On behalf of the team of authors we express our great pleasure for your attention to our work. We provide a new version 2 of the article with additions and new data, taking into account all comments from all reviews. Figures at the end.

Comments 1 “Oncolytic viruses” (L55) appear in the text without any further description or references. It should be stated that oncolytic viruses specifically replicate in tumor cells killing them while the normal cells are generally unaffected.

Response 1 - Totally agree added definition.

Oncolytic viruses (OVs) represent a novel class of cancer immunotherapy agents that preferentially infect and kill cancer cells and promote protective antitumor immunity [9]. 

Comments 2 - What is the statement “the therapeutic impact in in vivo experiments could be even more pronounced” (L490) based on? Compare this statement to “In vitro sensitivity of malignant tumor cells to an oncolytic agent does not consistently predict therapeutic efficacy in vivo” (L496), which contradicts the previous one!

Response 2 - We agree with your observation and have rephrased the conclusions.

We suppose these findings promising and probably in the case of human pancreatic carcinomas, where in vitro CPE 17D observed stronger, the therapeutic impact in vivo experiments could be even more pronounced. However, we recognize the risk that our expectations may not be met due to a number of factors by both the mechanism of cell resistance impairing viral tropism and the shortcomings of the mouse models for the 17D study.

Comments 3 - The authors discuss intratumoral administration as favorable to systemic delivery based on toxicity (L504) without even mentioning that typically oncolytic viruses replicate specifically in tumor cells and cause no or relatively little harm to normal tissue.

Response 3 - explained and links added

           Intratumoral administration of immunotherapeutic agents represents an effective strategy while minimizing systemic toxicity [29], and the fact that immunization does not reduce efficacy but, on the contrary, markedly increases it [13], providing an interesting safety feature, potentially enables the use of higher doses of 17D in therapy. Because the direct delivery of naked viruses through intravenous injection presents challenges, including rapid clearance by the immune system, inadequate accumulation in tumors, and significant side effects there is a need to develop new innovative strategies for delivering OVs [30].

Comments 4 - I am not sure the authors have understood the mechanism of Sindbis virus (a self-replicating RNA virus belonging to alphaviruses) infections (L537) as in the references [32, 33] replication-deficient vectors are used triggering bystander immunity.

Response 4 - We agree. The paragraph has been deleted for lack of relevance to our problem.

Comments 5 - The authors claim that the lack of superior therapeutic effect after repeated administration is due to “The observed lack of effect could be attributed to strong immune cell infiltration within the tumor” (L549) instead of simply considering the possible resistance against YFV after the first immunization.

Response 5 - absolutely accurate! In the new version, we are reviewing this problem.

3.8. Viral load, neutralizing antibodies, and white blood cells analysis

"The increased number of injections (x3) did not improve the effectiveness of the therapy. Each subsequent viral load caused an increase in the level of neutralizing antibodies in sera of the 17D-immunized mice (Fig. 9A). The highest antibody titer against 17D at the humane endpoint was detected in the group with triple virotherapy compared to the control group, where the titer started to decrease. Thus, we increased the resistance to 17D with each subsequent injection. This could have been avoided by increasing the dose of virus with each subsequent injection."

Comments 6 The statement “The live-attenuated yellow fever vaccine strain 17D has been in use for over ninety years” (L569) is incorrect. Although the YFV was discovered approximately 90 years ago, the 17D strain has been used for 70 years.

Response 6 Thank you, we've clarified. 

"The live-attenuated yellow fever vaccine strain 17D has been in use for 70 years and remains the gold standard for vaccines due to its exceptional immunogenicity. "

Specific comments - Many thanks for such detailed corrections. All of them are taken into account in the text.

The language is poor and requires a major revision. Response: I couldn't agree more, version 1 was very weak. English has been significantly improved in version 2.

Sincerely Y. Biryukova et al
